**eLife** SHORT REPORT

# Decoupling of the onset of anharmonicity between a protein and its surface water around 200 K

**Lirong Zheng**[1†‡], **Bingxin Zhou**[1,2*†], **Banghao Wu**[1,3†], **Yang Tan**[1,2], **Juan Huang**[1,3], **Madhusudan Tyagi**[4,5], **Victoria García Sakai**[6], **Takeshi Yamada**[7], **Hugh O'Neill**[8], **Qiu Zhang**[8], **Liang Hong**[1,2,9,10*]

[1]Institute of Natural Sciences, Shanghai Jiao Tong University, Shanghai, China; [2]Shanghai National Center for Applied Mathematics (SJTU Center), Shanghai Jiao Tong University, Shanghai, China; [3]School of Life Sciences and Biotechnology, Shanghai Jiao Tong University, Shanghai, China; [4]Department of Materials Science and Engineering, University of Maryland, College Park, United States; [5]NIST Center for Neutron Research, National Institute of Standards and Technology (NIST), Gaithersburg, United States; [6]ISIS Pulsed Neutron and Muon Source, Rutherford Appleton Laboratory, Science & Technology Facilities Council, Didcot, United Kingdom; [7]Neutron Science and Technology Center, Comprehensive Research Organization for Science and Society, Ibaraki, Japan; [8]Biology and Soft Matter Division, Oak Ridge National Laboratory, Oak Ridge, United States; [9]Zhangjiang Institute for Advanced Study, Shanghai Jiao Tong Univeristy, Shanghai, China; [10]Shanghai Artificial Intelligence Laboratory, Shanghai, China

**\*For correspondence:**
bingxin.zhou@sjtu.edu.cn (BZ); hongl3liang@sjtu.edu.cn (LH)

[†]These authors contributed equally to this work

**Present address:** [‡]Department of Cell and Developmental Biology & Michigan Neuroscience Institute, University of Michigan Medical School, Ann Arbor, United States

**Competing interest:** The authors declare that no competing interests exist.

## eLife assessment

The study answers the **important** question of whether the conformational dynamics of proteins are slaved by the motion of solvent water or are intrinsic to the polypeptide. The results from neutron scattering experiments, involving isotopic labelling, carried out on a set of four structurally different proteins are **convincing**, showing that protein motions are not coupled to the solvent. A strength of this work is the study of a set of proteins using spectroscopy covering a range of resolutions. The work is of broad interest to researchers in the fields of protein biophysics and biochemistry.

**Abstract** The protein dynamical transition at ~200 K, where the biomolecule transforms from a harmonic, non-functional form to an anharmonic, functional state, has been thought to be slaved to the thermal activation of dynamics in its surface hydration water. Here, by selectively probing the dynamics of protein and hydration water using elastic neutron scattering and isotopic labeling, we found that the onset of anharmonicity in the two components around 200 K is decoupled. The one in protein is an intrinsic transition, whose characteristic temperature is independent of the instrumental resolution time, but varies with the biomolecular structure and the amount of hydration, while the one of water is merely a resolution effect.

## Introduction

It is well established that the internal dynamics of a protein is crucial for its functions, including allosteric conformational changes (**Martin, 2001**), ligand binding (**Balog et al., 2004**) and enzymatic reactions

(*Hay and Scrutton, 2012*). In particular, hydrated proteins exhibit a dynamical transition around 200 K, across which the slope of the temperature dependence of the atomic displacements changes significantly and the biomolecule transforms from a rigid, harmonic state, to a flexible, anharmonic form (*Rupley and Careri, 1991*; *Vitkup et al., 2000*; *Rasmussen et al., 1992*; *Wood et al., 2008*; *Roh et al., 2005*; *Zaccai, 2000*; *Doster et al., 1989*; *Hong et al., 2013*; *Schiró et al., 2012*). Although exceptions have been reported (*Daniel et al., 1998*), the dynamical transition has been linked to the thermal onset of function in a number of proteins, for example, myoglobin (MYO; *Austin et al., 1975*), ribonuclease (*Rasmussen et al., 1992*), elastase (*Ding et al., 1994*), and bacteriorhodopsin (*Ferrand et al., 1993*), all of which become inactive below the dynamical transition temperature. The dynamical transition of protein has garnered various explanations. One theory suggests it is due to the behavior of water in the hydration shell, transitioning from rigid to fluid at certain temperatures, thus influencing protein flexibility (*Wood et al., 2008*; *Schiró et al., 2012*; *Fenimore et al., 2002*; *Frauenfelder et al., 2009*; *Qin et al., 2016*; *Lewandowski et al., 2015*). Another theory considers the transition as an inherent property of the protein, where thermal energy allows the protein to access a wider range of conformations (*Nickels et al., 2012*).

A prevailing scenario is that the internal dynamics of the protein is slaved to the motion of the surrounding hydration water (*Wood et al., 2008*; *Schiró et al., 2012*; *Fenimore et al., 2002*; *Frauenfelder et al., 2009*; *Qin et al., 2016*; *Lewandowski et al., 2015*), and thus the protein dynamical transition results from the changes in the dynamics of the hydration water with temperature (*Vitkup et al., 2000*; *Wood et al., 2008*; *Schiró et al., 2012*; *Frauenfelder et al., 2009*; *Tournier et al., 2003*). This scenario is indirectly supported by the experimental finding that the presence of the protein dynamical transition requires a minimum amount of hydration water, ~0.2 g water/g protein (*Rupley and Careri, 1991*; *Roh et al., 2005*). Further support comes from the results of all-atom molecular dynamics simulations, suggesting that it is the activation of the translational motions of surface water molecules around 200 K that leads to the dynamical transition in the underlying protein (*Vitkup et al., 2000*; *Wood et al., 2008*; *Tournier et al., 2003*).

This 'slaving' scenario can be examined directly by an experiment using isotopic labeling in combination with elastic neutron scattering methods (*Wood et al., 2008*; *Nickels et al., 2012*). Neutrons are highly sensitive to hydrogen atoms as their incoherent scattering cross section is an order of magnitude higher than the incoherent and coherent scattering cross sections of other elements (*Liu et al., 2017*; *Gaspar et al., 2010*; *Hong et al., 2014*). Thus, neutron signals collected on an ordinary protein powder hydrated in $D_2O$ reflect the dynamics of the protein while signals from the perdeuterated sample in $H_2O$ inform about the motion of water. The experimental results derived from this combined approach are, however, inconsistent (*Wood et al., 2008*; *Nickels et al., 2012*; *Benedetto, 2017*). Measurements performed on perdeuterated maltose-binding protein hydrated in $H_2O$ revealed a harmonic-to-anharmonic transition for hydration water taking place at the same temperature as that of the underlying protein (*Wood et al., 2008*). In contrast, a similar experiment on perdeuterated green fluorescence protein showed that the anharmonic onset in hydration water occurs at a lower temperature than that of the protein (*Nickels et al., 2012*). More recent measurements on lysozyme (LYS) hydrated in both $D_2O$ and $H_2O$ found that the transition temperature of protein and water coincided when examining their atomic displacements at 1 ns, but took place at different temperatures when changing the explored time scale to 3 ns (*Benedetto, 2017*). Therefore, there remains an unanswered question concerning whether the transition in dynamics of protein around 200 K is indeed

**Table 1.** Relative content of each secondary structure in the proteins.

| Protein | Lysozyme | Myoglobin | Cytochrome P450 | Green fluorescent protein |
|---|---|---|---|---|
| Abbreviation | LYS | MYO | CYP | GFP |
| PDB ID | 1AKI | 2V1K | 2ZAX | 1EMB |
| Alpha-helix* | 40% | 76% | 52% | 7% |
| Beta-sheet* | 12% | 0% | 11% | 50% |
| Loop and turn* | 48% | 24% | 37% | 43% |

*The relative content of each secondary structure is defined by mass fraction.

coupled to that of the hydration water, whose resolution is of fundamental importance to understand the mechanism governing the nature of their interaction.

To address this, it requires a systematic measurement of the temperature dependence of atomic displacements of the protein and its surface water separately, as a function of hydration levels, $h$ (g water/g protein), and at different time scales (instrument resolutions). Here, we performed elastic neutron scattering experiments on a number of protein powders hydrated in $D_2O$ and on the perdeuterated counterparts hydrated in $H_2O$, to track the dynamics of protein and hydration water independently. Moreover, using a range of neutron instruments with distinct resolutions, we tested the effect of the explored time scales on the dynamics of the two components. Four globular proteins with different secondary and tertiary structures (see *Figure 1—figure supplement 1* and *Table 1*) were studied here. We found that the onset temperature ($T_{on}$) of the protein dynamical transition varies with both biomolecular structure and hydration level, but is independent of the instrumental resolution time. Conversely, $T_{on}$ of the hydration water is insensitive to both the protein structure and the level of hydration, but solely determined by the instrument resolution. Therefore, the dynamical transition of the protein is decoupled from the onset of anharmonic dynamics of its hydration water around 200 K. The onset in water cannot be assigned to a physical transition, but to a resolution effect. In contrast, the protein dynamical transition is an intrinsic change in the dynamics of the biomolecules. Complementary differential scanning calorimetry (DSC) measurements revealed a step-like change in the heat flow around the transition temperature of the protein, similar to the glass transition observed in polymers. This suggests that the dynamical transition in the protein results from a similar process involving the freezing of the structural relaxation of the protein molecules beyond equilibrium.

## Results

### Elastic neutron scattering experiments

The quantity measured in the neutron experiment is the elastic intensity, that is the intensity of the elastic peak in the dynamic structure factor, $S(q, \Delta t)$, where $q$ is the scattering wave vector and $\Delta t$ is the resolution time of the neutron spectrometer. $S(q, \Delta t)$ is an estimate of the average amplitude of the atomic motions up to $\Delta t$ (*Hong et al., 2013*; *Liu et al., 2017*). Three neutron backscattering spectrometers were chosen to cover a wide range of time scales; HFBS at the NIST Center for Neutron Scattering, USA, DNA at the Materials and Life Science Experimental Facility at J-PARC in Japan, and OSIRIS at the ISIS Neutron and Muon Facility, UK. The instrumental energy resolutions are 1, 13, 25.4, and 100 µeV, corresponding to time scales of ~1 ns, ~80 ps, ~40 ps, and ~10 ps, respectively. Four globular proteins were investigated, MYO, cytochrome P450 (CYP), LYS, and green fluorescent protein (GFP), the detailed structural features of which are presented in *Figure 1—figure supplement 1* and *Table 1*. For simplicity, the hydrogenated and perdeuterated proteins are noted as H- and D-protein, respectively. Details of the sample preparation and neutron experiments are provided in Materials and Methods (*Table 2*).

### Dynamics of protein

*Figure 1a–g* shows the temperature dependence of $S(q, \Delta t)$ collected on H-LYS, H-MYO, and H-CYP in dry and hydrated state with $D_2O$ measured by neutron spectrometers of different resolutions, $\Delta t$. Since the measurements were performed on H-protein in $D_2O$, the signals reflect the dynamics of the proteins. A clear deviation can be seen in the temperature dependence of $S(q, \Delta t)$ for the hydrated protein from that of the dry powder, which is defined as the onset temperature, $T_{on}$ (*Roh et al., 2005*; *Schiró et al., 2012*; *Benedetto, 2017*; *Roh et al., 2006*; *Schirò et al., 2015*), of the protein dynamical transition. The advantage of such definition is that it highlights the effect of hydration on the anharmonic dynamics in proteins while removing the contribution from the local side groups, for example, methyl groups, whose motions are hydration independent (*Hong et al., 2013*; *Hong et al.,*

**Table 2.** The secondary structure content of cytochrome P450 (CYP) protein at different hydration levels.

| | Alpha-helix | Beta-sheet | Loop and turn |
|---|---|---|---|
| CYP ($h = 0.2$) | 52% | 11% | 37% |
| CYP ($h = 0.4$) | 52% | 11% | 37% |

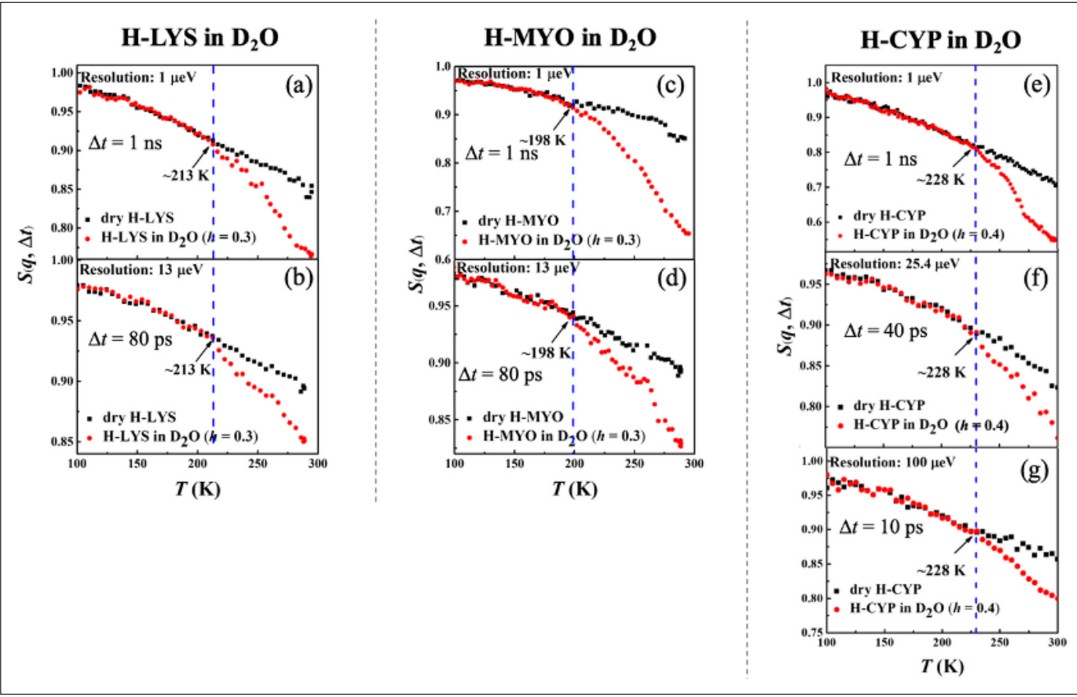

**Figure 1.** Resolution dependence of the onset of protein dynamical transition. Neutron spectrometers with different resolutions (1, 13, 25.4, and 100 µeV) were applied. Elatic intensity $S(q, \Delta t)$ of (**a, b**) dry H-LYS and H-LYS in $D_2O$ at $h = 0.3$, (**c, d**) dry H-MYO and H-MYO in $D_2O$ at $h = 0.3$, and (**e–g**) dry H-CYP and H-CYP in $D_2O$ at $h = 0.4$. All the experimental $S(q, \Delta t)$ are normalized to data measured at ~10 K and summed over values of $q$ ranging from 0.45 to 1.75 Å$^{-1}$. The dashed lines in each figure identify the onset temperatures of the transition, $T_{on}$, where the neutron data of the hydrated system deviate from the dry form.

The online version of this article includes the following figure supplement(s) for figure 1:

**Figure supplement 1.** Structures of proteins studied in this work.

**Figure supplement 2.** Resolution dependence of the onset of protein dynamical transition.

*2012*). As shown in *Schiró et al., 2012*; *Liu et al., 2018*, the activation temperature of the rotations of methyl group varies with the instrument resolution, which will cloud the present analysis. Two important conclusions can be drawn from *Figure 1* ($T_{on}$ is summarized in Table 5). (1) $T_{on}$ is distinct for each protein, LYS (213 K), MYO (198 K), and CYP (228 K), and (2) it is independent of the time scale explored even though the resolutions of the neutron spectrometers differ by orders of magnitude. Using the same set of data, we also analyzed the temperature dependence of the mean-squared atomic displacements, $<x^2(\Delta t)>$ (see results in *Figure 1—figure supplement 2*) and obtained similar conclusions. We further calculated $T_{on}$ of H-protein in $D_2O$ in the $q$-range from 0.45 to 0.9 and 1.1 to 1.75 Å$^{-1}$. As shown in *Table 3*, *Table 4*, and *Table 5*, the $q$-range does not alter the $T_{on}$ of proteins.

These findings suggest that the dynamical transition in the protein is an intrinsic property of the hydrated biomolecule, and it depends on the structure and chemistry of the protein concerned. Our results are consistent with *Schiró et al., 2012*, which demonstrated that $T_{on}$ in both protein and polypeptide is independent of the resolution of the neutron spectrometer, if one carefully removes the contributions from methyl rotations and vibrations to $<x^2(\Delta t)>$ measured by elastic neutron scattering. Additionally, *Benedetto, 2017* showed that, as compared to the dry form, the $D_2O$-hydrated LYS presents an approximately resolution-independent $T_{on}$, again in agreement with our findings.

*Figure 2* compares the temperature dependence of $S(q, \Delta t)$ measured on H-CYP and H-LYS in $D_2O$ at different hydration levels, $h$ ($T_{on}$ is summarized in *Table 6*). Evidently $T_{on}$ of the protein increases from 228 to 248 K when reducing $h$ from 0.4 to 0.2 (*Figure 2a*) by using the neutron instrument with $\Delta t = 1$ ns. A similar hydration dependence of $T_{on}$ is also observed when we replot the neutron data measured on H-LYS hydrated in $D_2O$ reported in *Roh et al., 2006* (*Figure 2c*). It can be found that $T_{on}$ of LYS changes systematically from 195 to 225 K, when decreasing $h$ from 0.45 to 0.18. Similar conclusion can be obtained when we analyzed $<x^2(\Delta t)>$ (see *Figure 2—figure supplement 1*). The

**Table 3.** $T_{on}$ of protein in $q$-ranges from $q$ = 0.45–0.9 Å$^{-1}$.

|  | 1 ns | 80 ps | 40 ps | 10 ps |
| --- | --- | --- | --- | --- |
| LYS | 213 K | 213 K | - | - |
| MYO | 198 K | 198 K | - | - |
| CYP | 228 K | - | 228 K | 228 K |

dynamical transition temperature in lipid membranes is higher when the membrane is dry (***Popova and Hincha, 2011***). We also studied the secondary structure content and tertiary structure of CYP protein at different hydration levels ($h$ = 0.2 and 0.4) through molecular dynamics simulation. As shown in ***Table 2*** and ***Figure 2—figure supplement 2***, the extent of hydration does not alter the protein secondary structure content and overall packing. Thus, this result suggests that water molecules have more influence on protein dynamics than on protein structure.

The results from the neutron scattering experiments suggest that the dynamical transition in proteins is an intrinsic property of the biomolecule and strongly depends on the amount of water surrounding it. Such an intrinsic transition can result either from a critical phase transition, for example, water to ice (***Wood et al., 2007***; ***Fitter et al., 1999***), or from freezing of the structural relaxation of the system beyond the equilibrium time (~100–1000 s) of the experiment, in analogy to the glass transition in polymers from rubbery state to the glass form (***Ngai, 2004***; ***Frick and Richter, 1995***; ***Frick et al., 1995***). Both of them will significantly increase the mechanical modulus of the material and suppress the atomic displacements at the fast time scales (pico-to-nanosecond) probed by the neutron spectrometers (***Wood et al., 2007***; ***Fitter et al., 1999***; ***Ngai, 2004***; ***Frick and Richter, 1995***; ***Frick et al., 1995***) like those used in this work. To explore the microscopic nature of the protein dynamical transition, we performed DSC measurements on CYP at dry, $h$ = 0.2 and 0.4. As illustrated in ***Figure 2b***, H-CYP at $h$ = 0.2 and $h$ = 0.4 exhibit a step-like transition in the heat flow at 245 and 225 K, respectively, while no such transition is observed in dry H-CYP. Such step-like transition in heat flow is normally defined as the glass transition in polymers (***Bassi et al., 2003***; ***Stolwijk et al., 2013***).

For simplicity, the step-like transition identified by DSC is noted as $T_{DSC}$. When comparing ***Figure 2b*** with ***Figure 2a***, one can find that the values of $T_{DSC}$ approximate those of $T_{on}$ probed by neutrons. $T_{DSC}$ of hydrated MYO was reported by literature to be 190 K (***Jansson and Swenson, 2010***), which is again in good agreement with the value of $T_{on}$ in ***Figure 1***. More importantly, $T_{DSC}$ and $T_{on}$ present the same hydration dependence, that is, both increase with decrease of $h$ (see ***Figure 2a, b***). Therefore, we can conclude that the onset of anharmonicity around 200 K in proteins measured by neutron scattering as shown in ***Figure 1*** results from the freezing of the structural relaxation of the biomolecule beyond the equilibrium when cooling the system below $T_{DSC}$, similar to the glass transition in polymers. Similar interpretation has also been suggested in ***Ngai et al., 2013***.

As the time scale probed by neutron spans from pico- to nanoseconds, it is too fast to allow us to directly 'see' structural relaxations of the protein around $T_{on}$. However, 'freezing' of the structural relaxation beyond the equilibrium time (~100–1000 s), that is, the measurement time of neutron experiments at each temperature, will turn the system into a 'frozen' solid form, which can significantly suppress the fast dynamics measured by neutron and cause the transition probed (***Ngai, 2004***; ***Frick and Richter, 1995***; ***Frick et al., 1995***; ***Ngai et al., 2013***). Moreover, water can be considered here as lubricant or plasticizer which facilitates the motion of the biomolecule (***Hong et al., 2012***; ***Roh et al., 2009***; ***Chen et al., 2018***). As widely observed in polymeric systems (***Verhoeven et al., 1989***; ***Zhang et al., 1999***; ***Cerveny et al., 2005***), adding water as plasticizer will significantly reduce the glass

**Table 4.** $T_{on}$ of protein in $q$-ranges from $q$ = 1.1–1.75 Å$^{-1}$.

|  | 1 ns | 80 ps | 40 ps | 10 ps |
| --- | --- | --- | --- | --- |
| LYS | 212 K | 213 K | - | - |
| MYO | 197 K | 199 K | - | - |
| CYP | 228 K | - | 227 K | 228 K |

**Table 5.** $T_{on}$ of protein at different time resolution.

|  | 1 ns | 80 ps | 40 ps | 10 ps |
|---|---|---|---|---|
| LYS ($h$ = 0.3) | 213 K | 213 K | - | - |
| MYO ($h$ = 0.3) | 198 K | 198 K | - | - |
| CYP ($h$ = 0.4) | 228 K | - | 228 K | 228 K |

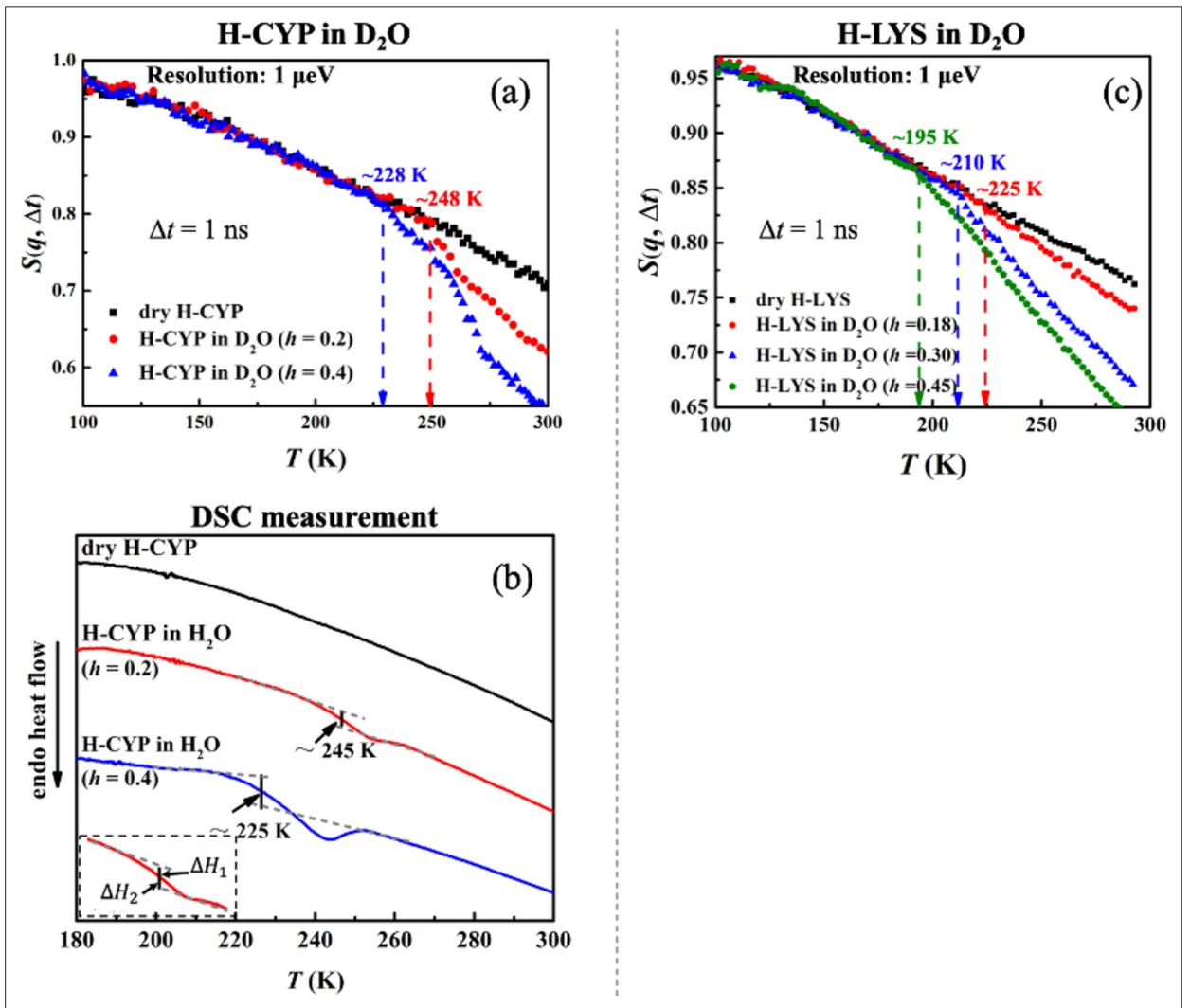

**Figure 2.** Hydration dependence of the onset of protein dynamical transition. $S(q, \Delta t)$ of (**a**) dry H-CYP and H-CYP in $D_2O$ at $h$ = 0.2 and 0.4 and (**c**) dry H-LYS and H-LYS in $D_2O$ at $h$ = 0.18, 0.30, and 0.45, all measured using HFBS with the instrumental resolution of 1 µeV. All the data in (**c**) were replotted from ***Roh et al., 2006***. (**b**) Differential scanning calorimetry (DSC) curves obtained for dry H-CYP and H-CYP in water at $h$ = 0.2 and 0.4. $T_{DSC}$ is defined as the midpoint between two heat flow baselines, where $\Delta H_1 = \Delta H_2$ (***Bassi et al., 2003***; ***Stolwijk et al., 2013***; ***ASTM International, 2014***).

The online version of this article includes the following figure supplement(s) for figure 2:

**Figure supplement 1.** Resolution dependence of the onset of protein dynamical transition.

**Figure supplement 2.** The three-dimensional (3D) structure of cytochrome P450 (CYP) protein at different hydration levels obtained from molecular dynamics (MD) simulations (PDB ID: 2ZAX).

**Figure supplement 3.** The potential energy as a function of MD trajectory time of cytochrome P450 (CYP).

**Table 6.** $T_{on}$ of protein at different hydration level.

|  | 0.18 | 0.2 | 0.3 | 0.4 | 0.45 |
|---|---|---|---|---|---|
| LYS (1 ns) | 225 K | - | 213 K | - | 195 K |
| CYP (1 ns) | - | 248 K | - | 228 K | - |
| CYP ($T_{DSC}$) | - | 245 K | - | 225 K | - |

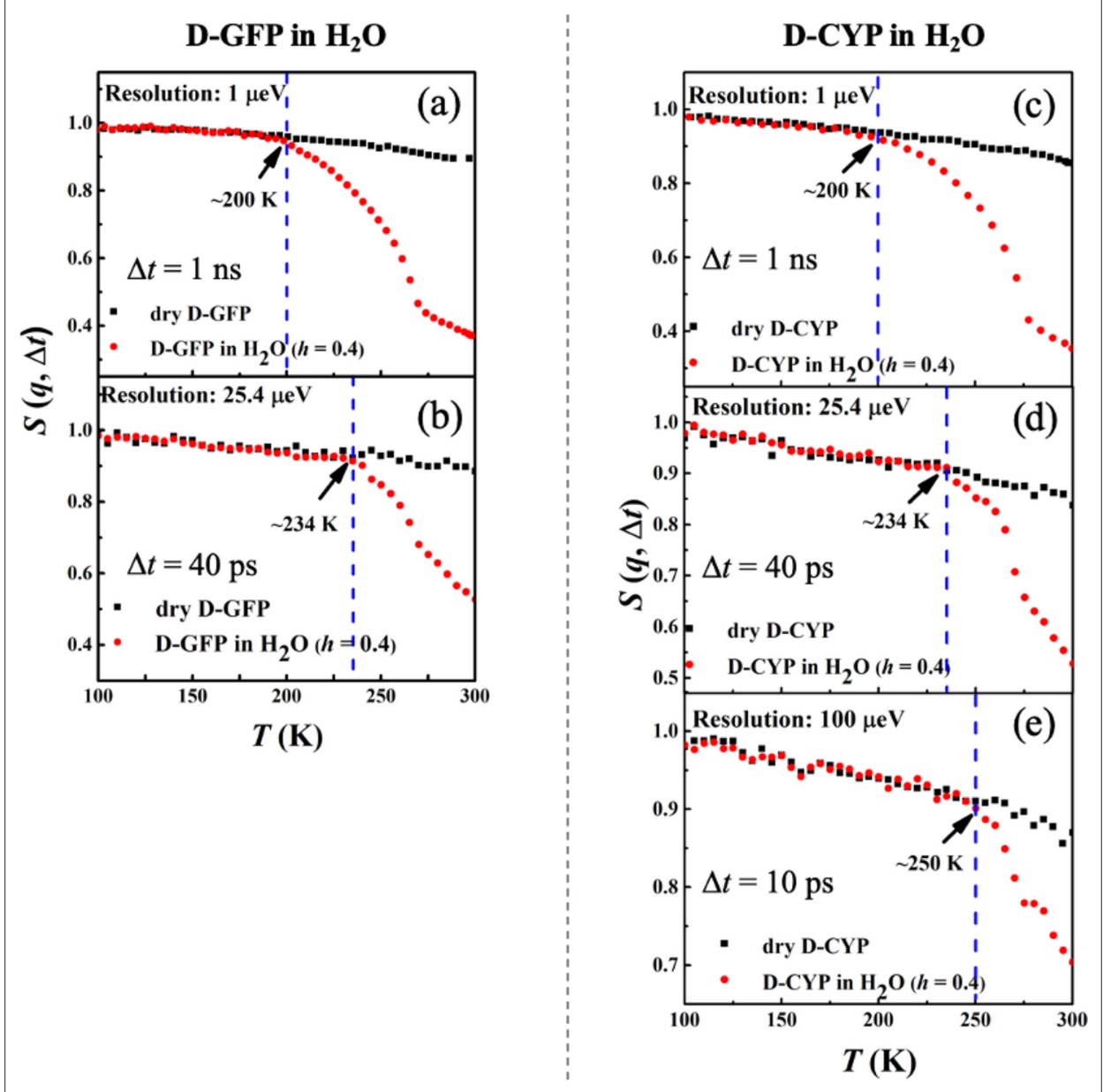

**Figure 3.** Resolution dependence of the anharmonic onset of hydration water. Neutron spectrometers with different resolutions (1, 25.4, and 100 µeV) were applied. $S(q, \Delta t)$ of (**a, b**) dry D-GFP and D-GFP in $H_2O$ at $h = 0.4$, and (**c–e**) dry D-CYP and D-CYP in $H_2O$ at $h = 0.4$.

The online version of this article includes the following figure supplement(s) for figure 3:

**Figure supplement 1.** Resolution dependence of the anharmonic onset of hydration water.

transition temperature of the polymers. This rationalizes the hydration effect on $T_{DSC}$ and $T_{on}$, both decreasing with increase of $h$.

## Dynamics in hydration water

*Figure 3a–e* shows the temperature dependence of $S(q, \Delta t)$ measured on perdeuterated GFP and CYP in dry and $H_2O$-hydrated forms. In these samples, the measured signal reflects primarily the motions of water molecules. Two important observations arise from the data. First, $T_{on}$ of hydration water for these two proteins strongly depends on the resolution of the spectrometer, increasing drastically from 200 to 250 K when reducing $\Delta t$ from 1 ns to 10 ps. Similar conclusions are obtained when we analyzed the temperature dependence of $<x^2(\Delta t)>$ (see *Figure 3—figure supplement 1*). The observation of a dependence of $T_{on}$ on $\Delta t$ is typical for a thermally activated process, which occurs when the characteristic relaxation time becomes comparable to the instrumental resolution, and the relaxation process is said to enter the time window of the instrument (*Liu et al., 2018*; *Schiró et al., 2010*). In this case, it means that the relaxation time, $\tau$, of hydration water is 10 ps at 250 K, 40 ps at 234 K, and 1 ns at 200 K. Assuming an Arrhenius-type process, $\tau = \tau_0 \exp\left(\frac{\Delta U}{k_B T}\right)$, the energy barrier $\Delta U$ can be estimated to be ~38 kJ/mol. Our results thus demonstrate that the anharmonic onset of the hydration water is in fact not a real physical transition but merely a resolution effect. It occurs as the relaxation time $\tau$ of water, which varies continuously with temperature, happens to cross the instrument resolution, $\Delta t$, on the pico-to-nanosecond time scales at $T_{on}$. Our findings agree with reports from dielectric measurements, the signal of which is highly sensitive to the rotation of hydration water (*Pawlus et al., 2008*; *Khodadadi et al., 2008*). They showed a smooth temperature dependence of the characteristic relaxation time in the range from 170 to 250 K without any sudden changes (*Pawlus et al., 2008*; *Khodadadi et al., 2008*). Moreover, our data also agree with *Doster et al., 2010*, which demonstrated that the characteristic relaxation time of protein-surface water, measured on $H_2O$-hydrated perdeuterated C-phycocyanin by quasi-elastic neutron scattering, changes smoothly over temperature without any disruptions around the dynamical transition temperature of the protein. Second, the onset temperature of the hydration water is independent of the protein structure when $\Delta t$ is fixed, since the values of both GFP and CYP are identical.

Furthermore, the hydration dependence of the anharmonic onset of the water is presented in *Figure 4*, which shows that $T_{on}$ remains constant with $h$ as long as the instrument resolution is fixed. This behavior is drastically different from that of the protein (*Figure 2*). The same conclusions can be obtained when analyzing $<x^2(\Delta t)>$ (see *Figure 4—figure supplement 1*).

## Conclusion and discussion

By combining elastic neutron scattering with isotopic labeling, we have been able to probe the dynamics of the protein and surface water separately, as a function of temperature, protein structural composition, hydration level, and time scale. We found that the anharmonic onsets of the two components around 200 K are clearly decoupled and different in origin. The protein shows an intrinsic transition, whose temperature depends on the structure of the protein and the hydration level, and not on the instrument used to measure it. It has a thermodynamic signature similar to the glass transition in polymers as confirmed by DSC, and thus can be assigned to the freezing of the structural relaxation of the protein beyond the experimental equilibrium time (100–1000 s). In contrast, the temperature at which the onset of anharmonicity happens in the hydration water is given by the instrument resolution, independent of both the biomolecular structure and the level of hydration.

Based on our findings, we can infer that, in some cases, the dynamical transition of a protein can coincide with the anharmonic onset of its surface water if one characterizes the system using a single-neutron instrument with a fixed resolution. But such coincidence will be torn apart if the measurements were conducted by using instruments of different resolutions or at different amounts of hydration, such as in the present work. This rationalizes the seemingly contradictive results reported in the literature (*Wood et al., 2008*; *Nickels et al., 2012*; *Benedetto, 2017*).

The protein dynamical transition has long been thought to connect to the thermal onset of the functionality of the biomolecule. Our experiments suggest that this transition in protein is an intrinsic property of the hydrated protein that its structural relaxation is activated upon heating above the onset temperature. This structural relaxation might be associated with conformational jumps of the biomolecules among different functional states, such as the states with the ligand-binding pocket

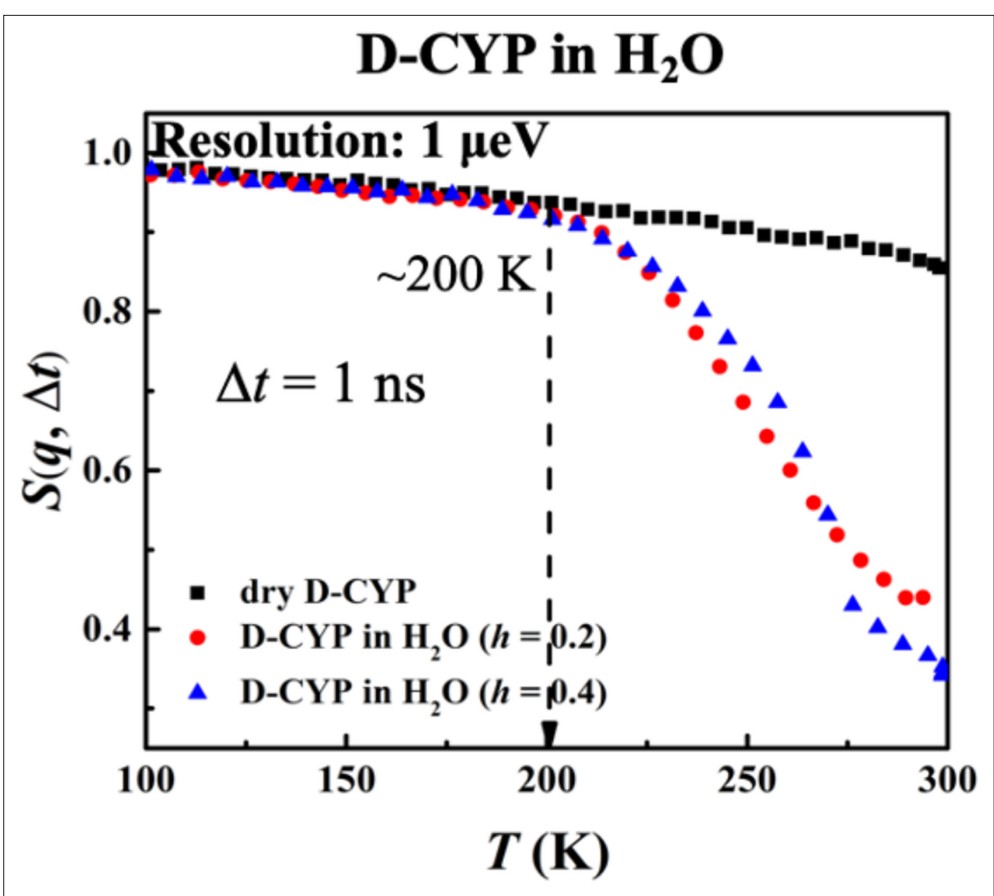

**Figure 4.** Hydration dependence of the anharmonic onset of hydration water. $S(q, \Delta t)$, for dry D-CYP and D-CYP in H$_2$O at $h$ = 0.2 and 0.4, measured using HFBS neutron instrument with an energy resolution of 1 μeV.

The online version of this article includes the following figure supplement(s) for figure 4:

**Figure supplement 1.** Hydration dependence of the anharmonic onset of hydration water.

being opened or closed. Unfreezing of the protein structural relaxation might facilitate these conformational jumps, turning on its functionality. However, as revealed by *Mamontov et al., 2010*, the denatured form of LYS also exhibits a dynamical transition, similar to that seen in its folded native form. Additionally, the dynamical transition also can be found in the mixture of amino acids (*Schiró et al., 2012*). Hence, one can argue that the activation of the structural relaxation of the biomolecule above the dynamical transition temperature is a necessary but insufficient condition for the protein to function, as the latter also requires the biomolecule assuming the correctly folded three-dimensional structure. The findings in this work help further the understanding of the microscopic mechanism governing the dynamics in proteins and their hydration water, as well as their interactions at the cryogenic temperature. More importantly, we demonstrated that the protein dynamical transition is a real transition, connecting to unfreezing of the biomolecular structural relaxation, which could be crucial for activating the function.

# Materials and methods

**Key resources table**

| Reagent type (species) or resource | Designation | Source or reference | Identifiers | Additional information |
|---|---|---|---|---|
| Strain, strain background (*Escherichia coli*) | *Escherichia coli*, BL21(DE3) | Sigma-Aldrich | CMC0016 | |

*Continued on next page*

*Continued*

| Reagent type (species) or resource | Designation | Source or reference | Identifiers | Additional information |
|---|---|---|---|---|
| Peptide, Recombinant protein | Lysozyme, chicken egg white | Sigma-Aldrich | CAS: 12650-88-3 | |
| Peptide, Recombinant protein | Myoglobin, equine skeletal muscle | Sigma-Aldrich | CAS: 100684-32-0 | |
| Chemical compound, drug | $H_2O$ | Millipore | | |
| Chemical compound, drug | $D_2O$ | Sigma-Aldrich | CAS: 7789-20-0 | |

## Sample preparation

Hydrogenated LYS from chicken egg white and hydrogenated MYO from equine skeletal muscle were purchased from Sigma-Aldrich (Shanghai, China). The expression and purification of hydrogenated and perdeuterated CYP (we used P450 from *Pseudomonas putida* for the study) and GFP are described previously *Liu et al., 2017*. In order to exclude the effect of ions, the proteins were dialyzed before experiments. For simplification, the hydrogenated protein and perdeuterated protein are denoted as H- and D-proteins in the manuscript, respectively. All the H-proteins were dissolved in $D_2O$ to allow full deuterium exchange of all exchangeable hydrogen atoms and then lyophilized for 12 hr to obtain the dry sample. The lyophilized H-protein is then put into a desiccator with $D_2O$, placed in the glove box purged with nitrogen gas, to absorb $D_2O$ till the desired hydration level, $h$ (g water/g protein). In contrast, the preparation of the deuterated proteins was conducted in the opposite way. The D-proteins were dissolved in $H_2O$ to allow full hydrogen exchange of all exchangeable deuterium atoms and then lyophilized for 12 hr to obtain the dry sample. The lyophilized D-protein is then put into a desiccator with $H_2O$ to absorb $H_2O$ till the desired $h$. The ultrapure water ($H_2O$) was supplied by a Millipore Direct-Q system (18.2 MΩ cm at 25°C). The deuterium oxidized ($D_2O$, 99.9 atom % D) was purchased from Sigma-Aldrich (Shanghai, China). The hydration levels of protein samples were controlled by measuring the sample weights before and after water adsorption. In this work, $h$ ranges from 0.02 (lyophilized dry form), 0.2, 0.3–0.4, where $h = 0.4$ corresponds to a case that roughly a single layer of water molecules covers the protein's surface. The dry H-CYP, H-LYS, H-MYO, and their $D_2O$-hydrated forms at $h = 0.2$, 0.3, or 0.4, and the dry D-GFP and D-CYP, and their $H_2O$-hydrated powders at $h = 0.4$ are prepared for neutron scattering experiments. The accuracy of $h$ is controlled within 10% error. For example, $h = 0.4 \pm 0.04$ g water/g protein. All samples were sealed tightly in the aluminum cans in nitrogen before the neutron scattering experiments.

The dry H-CYP lyophilized in $H_2O$ and the ones hydrated in $H_2O$ at $h = 0.2$ and 0.4 are prepared for the DSC measurement.

## Elastic incoherent neutron scattering

The elastic scattering intensity $S(\boldsymbol{q}, \Delta t) \approx I_{inc}(\boldsymbol{q}, \Delta t) = \frac{1}{N} \sum_{j}^{N} b_{j,inc}^2 \langle \exp[-i\boldsymbol{q} \cdot r_j(0)] \exp[i\boldsymbol{q} \cdot \boldsymbol{r}_j(\Delta t)] \rangle$ is

normalized to the lowest temperature (~10 K) and is approximately the value of the intermediate scattering function when decaying to the instrument resolution time, $\Delta t$. All the $S(q, \Delta t)$ was obtained in the temperature range of ~10–300 K during heating process with the rate of 1.0 K/min by using the HFBS at NIST, DNA at J-PARC, and OSIRIS at ISIS. The energy resolutions of HFBS, DNA, and OSIRIS are 1, 13, 25.4, and 100 µeV, corresponding to the resolution times of ~1 ns, ~80 ps, ~40 ps, and ~10 ps, respectively. The results from instruments with various resolutions were summed over the same $q$ from 0.45 to 1.75 Å$^{-1}$.

## Differential scanning calorimetry

DSC measurements were performed by using the METTLER instruments DSC3+. The sample was sealed in a pan of aluminum. An empty pan was used as a reference. All the experiments were carried out in the temperature ranged from 150 to 300 K with heating rate of 1 K/min. The heating rate of DSC is the same as neutron experiments.

### Estimation of the mean-squared atomic displacement

The mean-squared atomic displacement $< x^2(\Delta t) >$ was estimated by performing Gaussian approximation, where $S(q, \Delta t) = \exp(-\frac{1}{6}q^2 < x^2(\Delta t) >)$. The values of $q$ used for Gaussian fitting ranges from 0.45 to 0.9 Å$^{-1}$.

## Protein samples used for experiments

We studied four globular proteins, MYO, CYP, LYS, and GFP, the detailed structural features of which are presented in *Figure 1—figure supplement 1* and *Table 1*. The four proteins differ significantly in both secondary and tertiary structures. MYO is primarily a helix protein while GFP is dominated by beta sheets. Moreover, LYS contains two structural domains linked by a hinge while the other three are single-domain proteins.

## Molecular dynamics simulation

The initial structure of protein CYP for simulations was taken from PDB crystal structure (2ZAX). Two protein monomers were filled in a cubic box. 1013 and 2025 water molecules were inserted into the box randomly to reach a mass ratio of 0.2 and 0.4 g water/1 g protein, respectively, which mimics the experimental condition. Then 34 sodium counter ions were added to keep the system neutral in charge. The CHARMM 27 force field in the GROMACS package was used for CYP, whereas the TIP4P/Ew model was chosen for water. The simulations were carried out at a broad range of temperatures from 360 to 100 K, with a step of 5 K. At each temperature, after the 5000 steps energy-minimization procedure, a 10-ns NVT (substance, volume, temperature) is conducted. After that, a 30-ns NPT (substance, pressure, temperature) simulation was carried out at 1 atm with the proper periodic boundary condition. As shown in *Figure 2—figure supplement 3*, 30 ns is sufficient to equilibrate the system. The temperature and pressure of the system ar controlled by the velocity rescaling method and the method by Parrinello and Rahman, respectively. All bonds of water in all the simulations were constrained with the LINCS (Linear Constraint Solver) algorithm to maintain their equilibration length. In all the simulations, the system was propagated using the leap-frog integration algorithm with a time step of 2 fs. The electrostatic interactions were calculated using the Particle Mesh Ewalds method. A non-bond pair-list cutoff of 1 nm was used and the pair-list was updated every 20 fs. All MD simulations were performed using GROMACS 4.5.1 software packages.

## Acknowledgements

This work is supported by the National Natural Science Foundation of China (11974239; 62302291), the Innovation Program of Shanghai Municipal Education Commission (2019-01-07-00-02-E00076). HO'N and QZ acknowledge the support of Center for Structural Molecular Biology (FWPERKP291) funded by the U.S. Department of Energy (DOE) Office of Biological and Environmental Research.

## Additional information

### Funding

| Funder | Grant reference number | Author |
|---|---|---|
| National Natural Science Foundation of China | 11974239 | Liang Hong |
| National Natural Science Foundation of China | 62302291 | Bingxin Zhou |
| Innovation Program of Shanghai Municipal Education Commission | 2019-01-07-00-02-E00076 | Liang Hong |
| Center for Structural Molecular Biology | FWPERKP291 | Hugh O'Neill Qiu Zhang |

The funders had no role in study design, data collection, and interpretation, or the decision to submit the work for publication.

## Author contributions
Lirong Zheng, Conceptualization, Data curation, Formal analysis, Supervision, Validation, Investigation, Visualization, Methodology, Writing – original draft, Writing – review and editing; Bingxin Zhou, Conceptualization, Funding acquisition, Validation, Investigation, Visualization, Methodology, Writing – original draft, Project administration, Writing – review and editing; Banghao Wu, Formal analysis, Validation; Yang Tan, Juan Huang, Formal analysis; Madhusudan Tyagi, Victoria García Sakai, Takeshi Yamada, Hugh O'Neill, Qiu Zhang, Resources; Liang Hong, Resources, Supervision, Funding acquisition, Writing – review and editing

## Author ORCIDs
Lirong Zheng https://orcid.org/0000-0001-6803-5048
Bingxin Zhou https://orcid.org/0000-0002-3897-9766
Liang Hong http://orcid.org/0000-0003-0107-336X

Reviewer #1 (Public Review): https://doi.org/10.7554/eLife.95665.4.sa1
Reviewer #2 (Public Review): https://doi.org/10.7554/eLife.95665.4.sa2
Author response https://doi.org/10.7554/eLife.95665.4.sa3

---

# Additional files

## Supplementary files
MDAR checklist

## Data availability
All the experimental and computational data are shown in main text. Access to the HFBS was provided by the Center for High-Resolution Neutron Scattering, a partnership between the National Institute of Standards and Technology and the National Science Foundation under Agreement No. DMR-1508249. The neutron experiment at the Materials and Life Science Experimental Facility of the J-PARC was performed under a user program (Proposal No. 2019A0020). We thank STFC for access to neutron scattering facilities at RB1800112. The original data are accessible via data cite: https://doi.org/10.5286/ISIS.E.RB1800112.

The following previously published dataset was used:

| Author(s) | Year | Dataset title | Dataset URL | Database and Identifier |
|---|---|---|---|---|
| Hong L, Sakai VG, Liu Z, Yang C | 2021 | Decoupling effect between protein and water | https://doi.org/10.5286/ISIS.E.95670743 | STFC ISIS Neutron and Muon Source, 10.5286/ISIS.E.RB1800112 |

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
