## [Editor Report · eLife assessment]

The study answers the **important** question of whether the conformational dynamics of proteins are slaved by the motion of solvent water or are intrinsic to the polypeptide. The results from neutron scattering experiments, involving isotopic labelling, carried out on a set of four structurally different proteins are **convincing**, showing that protein motions are not coupled to the solvent. A strength of this work is the study of a set of proteins using spectroscopy covering a range of resolutions. The work is of broad interest to researchers in the fields of protein biophysics and biochemistry.

---

## [Referee Report · Reviewer #1 (Public Review)]

Zheng et al. study the 'glass' transitions that occurs in proteins at ca. 200K using neutron diffraction and differential isotopic labeling (hydrogen/deuterium) of the protein and solvent. To overcome limitations in previous studies, this work is conducted in parallel with 4 proteins (myoglobin, cytochrome P450, lysozyme and green fluorescent protein) and experiments were performed at a range of instrument time resolutions (1ns - 10ps). The author's data looks compelling, and suggests that transitions in the protein and solvent behavior are not coupled and contrary to some previous reports, the apparent water transition temperature is a 'resolution effect'; i.e. instrument response is limited. This is likely to be important in the field, as a reassessment of solvent 'slaving' and the role of the hydration shell on protein dynamics should be reassessed in light of these findings.

---

## [Referee Report · Reviewer #2 (Public Review)]

Summary:

The manuscript entitled "Decoupling of the Onset of Anharmonicity between a Protein and Its Surface Water around 200 K" by Zheng et al. presents a neutron scattering study trying to elucidate if at the dynamical transition temperature water and protein motions are coupled. The origin of the dynamical transition temperature has been debated for decades, specifically its relation to hydration.

The study is rather well conducted, with a lot of effort to acquire the perdeuterated proteins, and some results are interesting.

---

## [Author Response]

The following is the authors’ response to the previous reviews.

**eLife assessment**
The study answers the important question of whether the conformational dynamics of proteins are slaved by the motion of solvent water or are intrinsic to the polypeptide. The results from neutron scattering experiments, involving isotopic labelling, carried out on a set of four structurally different proteins are convincing, showing that protein motions are not coupled to the solvent. A strength of this work is the study of a set of proteins using spectroscopy covering a range of resolutions. A minor weakness is the limited description of computational methods and analysis of data. The work is of broad interest to researchers in the fields of protein biophysics and biochemistry.

We thank the editors and reviewers for the positive and encouraging comments.

**Public Reviews:**

**Reviewer #1 (Public Review):**
Summary:Zheng et al. study the 'glass' transitions that occurs in proteins at ca. 200K using neutron diffraction and differential isotopic labeling (hydrogen/deuterium) of the protein and solvent. To overcome limitations in previous studies, this work is conducted in parallel with 4 proteins (myoglobin, cytochrome P450, lysozyme and green fluorescent protein) and experiments were performed at a range of instrument time resolutions (1ns - 10ps). The author's data looks compelling, and suggests that transitions in the protein and solvent behavior are not coupled and contrary to some previous reports, the apparent water transition temperature is a 'resolution effect'; i.e. instrument response limited. This is likely to be important in the field, as a reassessment of solvent 'slaving' and the role of the hydration shell on protein dynamics should be reassessed in light of these findings.Strengths:The use of multiple proteins and instruments with a rate of energy resolution/ timescales.

We thank the reviewer for highlighting our key findings.

Weaknesses:The paper could be organised to better allow the comparison of the complete dataset collected. The extent of hydration clearly influences the protein transition temperature. The authors suggest that "water can be considered here as lubricant or plasticizer which facilitates the motion of the biomolecule." This may be the case, but the extent of hydration may also alter the protein structure.

Following the reviewer’s suggestion, we studied the secondary structure content and tertiary structure of CYP protein at different hydration levels (h = 0.2 and 0.4) through molecular dynamics simulation. As shown in Table S2 and Fig. S6, the extent of hydration does not alter the protein secondary structure content and overall packing. Thus, this result also suggests that water molecules have more influence on protein dynamics than on protein structure. We added the above results in the revised SI.

**Reviewer #2 (Public Review):**
Summary:The manuscript entitled "Decoupling of the Onset of Anharmonicity between a Protein and Its Surface Water around 200 K" by Zheng et al. presents a neutron scattering study trying to elucidate if at the dynamical transition temperature water and protein motions are coupled. The origin of the dynamical transition temperature is highly debated since decades and specifically its relation to hydration.Strengths:The study is rather well conducted, with a lot of efforts to acquire the perdeuterated proteins, and some results are interesting.

We thank the reviewer for highlighting our key findings.

Weaknesses:The MD data presented appears to be missing description of the methods used.If these data support the authors claim that different levels of hydration do not affect the protein structure, careful analysis of the MD simulation data should be presented that show the systems are properly equilibrated under each condition. Additionally, methods are needed to describe the MD parameters and methods used, and for how long the simulations were run.

We have now added the methods of MD simulation into the revised SI.

“The initial structure of protein cytochrome P450 (CYP) for simulations was taken from PDB crystal structure (2ZAX). Two protein monomers were filled in a cubic box. 1013 and 2025 water molecules were inserted into the box randomly to reach a mass ratio of 0.2 and 0.4 gram water/1 gram protein, respectively, which mimics the experimental condition. Then 34 sodium counter ions were added to keep the system neutral in charge. The CHARMM 27 force field in the GROMACS package was used for CYP, whereas the TIP4P/Ew model was chosen for water. The simulations were carried out at a broad range of temperatures from 360 K to 100 K, with a step of 5 K. At each temperature, after the 5000 steps energy-minimization procedure, a 10 ns NVT is conducted. After that, a 30 ns NPT simulation was carried out at 1 atm with the proper periodic boundary condition. As shown in Fig. S7, 30 ns is sufficient to equilibrate the system. The temperature and pressure of the system is controlled by the velocity rescaling method and the method by Parrinello and Rahman, respectively. All bonds of water in all the simulations were constrained with the LINCS algorithm to maintain their equilibration length. In all the simulations, the system was propagated using the leap-frog integration algorithm with a time step of 2 fs. The electrostatic interactions were calculated using the Particle Mesh Ewalds (PME) method. A non-bond pair-list cutoff of 1 nm was used and the pair-list was updated every 20 fs. All MD simulations were performed using GROMACS 4.5.1 software packages.”

**Recommendations for the authors:**

**Reviewer #1 (Recommendations For The Authors):**
Response to author's changes:See public review: The MD data presented appears to be missing description of the methods used.If these data support the authors claim that different levels of hydration do not affect the protein structure, careful analysis of the MD simulation data should be presented that show the systems are properly equilibrated under each condition. Additionally, methods are needed to describe the MD parameters and methods used, and for how long the simulations were run.

We have now added the methods of MD simulation into the revised SI. Please see Reply 5.

**Reviewer #2 (Recommendations For The Authors):**
The authors answered my questions and substantially improved the manuscript.

We thank the reviewer for the encouraging comments .